# Enhancing Underwater Images via Asymmetric Multi-Scale Invertible Networks

Yuhui Quan*
South China University of Technology
Guangzhou, China
csyhquan@scut.edu.cn

Xiaoheng Tan
South China University of Technology
Guangzhou, China
csxiaohengtan@foxmail.com

Yan Huang†
South China University of Technology
Guangzhou, China
aihuangy@scut.edu.cn

Yong Xu‡
South China University of Technology
Guangzhou, China
yxu@scut.edu.cn

Hui Ji
National University of Singapore
Singapore
matjh@nus.edu.sg

## Abstract

Underwater images, often plagued by complex degradation, pose significant challenges for image enhancement. To address these challenges, the paper redefines underwater image enhancement as an image decomposition problem and proposes a deep invertible neural network (INN) that accurately predicts both the latent image and the degradation effects. Instead of using an explicit formation model to describe the degradation process, the INN adheres to the constraints of the image decomposition model, providing necessary regularization for model training, particularly in the absence of supervision on degradation effects. Taking into account the diverse scales of degradation factors, the INN is structured on a multi-scale basis to effectively manage the varied scales of degradation factors. Moreover, the INN incorporates several asymmetric design elements that are specifically optimized for the decomposition model and the unique physics of underwater imaging. Comprehensive experiments show that our approach provides significant performance improvement over existing methods.

## CCS Concepts

• **Computing methodologies → Computational photography**; **Image processing**; **Computational photography**.

## Keywords

Underwater image enhancement, Invertible neural networks, Multi-scale processing, Image recovery

**ACM Reference Format:**
Yuhui Quan, Xiaoheng Tan, Yan Huang, Yong Xu, and Hui Ji. 2024. Enhancing Underwater Images via Asymmetric Multi-Scale Invertible Networks.

*Yuhui Quan is also with Pazhou Lab, Guangzhou, China.
†Corresponding author: Yan Huang.
‡Yong Xu is also with Pazhou Lab, Guangzhou, China, and with Guangdong Provincial Key Laboratory of Multimodal Big Data Intelligent Analysis, Guangzhou, China.

In *Proceedings of the 32nd ACM International Conference on Multimedia (MM '24), October 28-November 1, 2024, Melbourne, VIC, Australia.* ACM, New York, NY, USA, 10 pages. https://doi.org/10.1145/3664647.3681098

## 1 Introduction

Underwater imaging is a challenging image acquisition task due to many factors such as water turbidity, scattering, absorption, and poor lighting. These factors can significantly degrade the quality of the captured images, resulting in various types of degradation such as low contrast, color distortion, haze effects, and blurring. Underwater image enhancement (UIE) aims to mitigate these issues, producing clearer and more visually appealing images. UIE has played a vital role in underwater exploration, research, surveillance, and engineering, with a broad spectrum of applications including oceanography, archaeology, marine biology, as well as underwater vehicles and robotics; see *e.g.* [24, 46, 59].

Owing to the degradation complexity in underwater environments, conventional UIE methods (*e.g.* [1, 2, 5, 6, 10, 13, 15, 18, 26, 39, 48, 61, 65–67, 69, 70]) typically focus on handling specific degradation effects often seen in digital photography, such as dehazing [10, 33, 61], color balancing [5, 13, 65, 67], and deblurring [61]. These methods often employ handcrafted models or predefined rules, which may be too simplistic to effectively address the complex degradation effects encountered in real-world underwater scenes, limiting their performance.

Recently, deep learning has become the dominant tool for UIE; see *e.g.* [8, 9, 12, 16, 17, 21, 22, 24, 25, 28–30, 35, 41, 43, 49, 50, 55, 58, 59]. Deep learning utilizes neural network (NN)-based models trained on datasets to address the complexity inherent in underwater images. While deep learning-based methods have surpassed traditional ones in UIE, there remains considerable room for performance improvement, especially in practical, real-world scenarios with diverse degradation factors and limited training data. Towards this end, we propose a deep NN called AMSIN (Asymmetric Multi-Scale Invertible Network).

### 1.1 Main Idea

***Recasting UIE as a decomposition problem:*** Our AMSIN is developed by reinterpreting UIE as an image decomposition task. An underwater image is understood as a result of a complex, non-linear process involving two key elements: a distortion-free image and

a distortion map. This map encapsulates all degradation characteristics of the underwater image, enabling the model to perfectly recreate the original underwater image by working with the latent image. With this reformulation, the NN needs to predict these two components from the underwater image, thus being inherently tasked with understanding the underlying degradation model to accurately infer the latent image. This perspective is crucial for effectiveness, making the NN more competently manage complex degradation effects typically found in practical underwater imagery, thereby improving the quality of reconstructed latent images.

*Implicit regularization via INN-based decomposition for UIE:*
The decomposition-based reformulation for UIE holds promise, but also introduces challenges on NN training. The training data only contains underwater and ground-truth (GT) images, but without distortion maps. Together with the absence of explicit composition models, it is likely to cause potential model overfitting [21, 35]. To mitigate the over-fitting issue in a decomposition-based paradigm, we introduce an INN architecture. This architecture is pivotal as it guarantees no loss of information during the feature extraction and disentanglement processes for both the latent image and the distortion map. Such a property ensures that the perfect reconstruction of the original underwater image from the predicted distortion map and latent image at each stage. Therefore, it implicitly imposes the reconstruction constraint inherent in the image decomposition model, effectively introducing extra regularization during training.

*Multi-scale invertible processing:* Considering the diverse scales of degradation factors inherent in underwater images, it is important to tackle underwater degradation across multiple scales. Thus, our AMSIN is designed with a multi-scale structure, featuring specialized encoder and decoder blocks for cross-scale representation. Each block consists of a series of invertible coupling layers and an invertible scaling module, maintaining dual feature flows for the latent image layer and the degradation layer. We also implement short pathways using split and concatenation operators between the encoder and decoder, which significantly enhances the transmission of multi-scale features. This multi-scale approach is also important for latent image estimation.

*Asymmetric structure:* In the encoder (decoder) blocks of AMSIN, we asymmetrically employ pixel unshuffling and Haar wavelet transform for the feature flows of latent image layer and the distortion map layer, respectively. This asymmetric design is tailored for enhanced spatial and frequency analysis, aiding in addressing the complex mixed degradation in an underwater image. Notably, for the latent image layer, we place greater emphasis on frequency information using Haar-wavelet-based features. In addition, we also asymmetrically define the reconstruction loss at different scales to improve gradient propagation during training, aligning with the asymmetric NN structure.

Another asymmetry is on the treatment on color channels. Specifically, we replicate the red channel of the degraded image for three times as the additional input and supervise it in training. This scheme emphasizes the recovery of red channel. It is motivated by the fact that the red channel of an underwater image usually experiences more severe degradation compared to the green and blue ones, due to the selective absorption of light in water.

## 1.2 Contributions

To summarize, the main contributions of this work include:

- The UIE is reformulated as an image decomposition problem, better aligning the NN design for UIE to effectively handle complex degradation in underwater images.
- We leverage INNs to implement the decomposition process of UIE, imposing the inherent perfect reconstruction constraint of decomposition for necessary regularization, particularly in the absence of GT degradation maps.
- We introduce a multi-scale structure to the INN for UIE, tailored for addressing the diverse scales of degradation factors in underwater images.
- We have integrated several asymmetric design elements optimized for the decomposition model and the physics of underwater imaging.

Extensive experiments demonstrate that our proposed AMSIN provides noticeable performance improvement over existing deep learning-based UIE methods.

## 2 Related Work

### 2.1 Conventional Methods for UIE

Conventional methods are roughly classified as model-free or model-driven, based on their use of physical models.

Model-driven methods leverage a physical model of underwater imaging, typically estimating parameters with certain priors and inverting the model for UIE. Due to the similarities between underwater scenes and atmospheric haze in terms of light scattering and absorption, atmospheric image formation models are commonly adapted, supplemented by various priors for regularization. Examples include the dark channel prior and its generalizations [10, 11, 32, 39, 48], red channel prior [61], hazy line prior [6], attenuation curve prior [57], and illumination channel sparsity prior [18]. Motivated by that light underwater attenuates with wavelength dependence [3], Akkaynak and Treirbitz [1] proposed a refined underwater imaging model and applied it to UIE [2].

Model-free methods adjust pixel values without leveraging physical models. The contrast enhancement methods [15, 67] improve image contrast through specific rules. The Retinex-based methods [13, 65, 69, 70] enhance separate illumination and reflectance layers from Retinex decomposition. The fusion-based methods [4, 5, 26, 66] combine differently enhanced images into one result.

### 2.2 Deep Learning Methods for UIE

Deep learning-based UIE methods primarily vary in their NN design. Convolutional NNs (CNNs) are widely used due to their effective local feature extraction; see [22, 28–30, 43, 44, 49, 50, 58, 59]. Wu *et al.* [59] introduced a two-stage CNN for sequential enhancement and refinement. Li *et al.* [28] developed a CNN featuring a multi-color space channel-attentive encoder and a medium transmission-guided decoder. Huo *et al.* [22] proposed a wavelet-enhanced multi-stage CNN for progressive refinement. Mu *et al.*[43] utilized a three-branch CNN to leverage multi-domain cues. Qi *et al.* [49] introduced a CNN with semantic attention and multi-scale perception. Mu *et al.* [44] proposed a CNN structure with dynamic convolutions and multi-scale design.

Generative adversarial networks (GANs) are also popular, with the capability of exploiting unpaired training data; see [8, 12, 17, 24, 25, 35, 41, 55]. Fabbri *et al.* [12] applied a CycleGAN. Guo *et al.* [17] proposed a multi-scale GAN. Islam *et al.* [24] developed a lightweight GAN for real-time UIE. Chen *et al.* [8] integrated object detection to enhance CycleGAN-based UIE. Liu *et al.* [41] presented a twin GAN with contrastive learning. Jiang *et al.* [25] introduced joint global and local discriminators.

Beyond CNNs and GANs, Chi *et al.* [9] proposed a gradient-guided Swin Transformer for global information exploitation. Guo *et al.* [16] developed a quality ranker for underwater images to enhance the performance of deep UIE models. Addressing the challenge of limited paired training data, in the context of UIE, various studies have been done on weakly-supervised color transfer learning [31, 36], semi-supervised learning [21, 51], unsupervised end-to-end training [14], and zero-shot self-supervised learning [27].

### 2.3 INNs for Image Restoration/Enhancement

INNs have been used in diverse image restoration and enhancement tasks, *e.g.*, super-resolution [37, 38, 60], denoising [19, 20, 42], desnowing [53], demoiréing [52], relighting [64], and raw reconstruction [62]. For instance, Lie *et al.* [37] implemented INNs for super-resolution with multi-scale analysis within a coupling block using varying convolution kernel sizes. In contrast, our approach conducts multi-scale analysis across coupling blocks, leading to greater efficiency.

The study of INN-based UIE is still scarce. A very recent study [68] introduced an INN-based conditional normalization flow for UIE. Our work differs from [68] in two significant perspectives. Firstly, we conceptualize UIE as a decomposition problem, unlike the conditional generation process in [68], leading to distinct roles for INNs in both studies. Secondly, our model employs an explicit multi-scale structure, optimizing efficient cross-scale processing of various degradation factors in underwater images.

## 3 Proposed Method

### 3.1 Overall Framework

Our proposed AMSIN aims at decomposing an underwater image $Y$ with distortion into a distortion-free image $X$ and a distortion map $D$ via a forward pass. The formation of $Y$ is totally dependent on $(X, D)$ through a complex and unknown composition process $f: Y = f(X, D)$. Then UIE can be done by the inverse $f^{-1}$. In our approach, these composition and decomposition models are simultaneously learned by the AMSIN with an INN structure.

See Fig. 1 for the outline of AMSIN. To fit the input-output dimension consistency required in an INN, the AMSIN replicates the red channel of the degraded image for three times, denoted by $\mathrm{RC}_3(Y)$, as the additional input. emphasizing the recovery of red-channel intensities in UIE. Specifically, let $\mathcal{G}$ denote the AMSIN and we have then:

Forward mode: $\quad \mathcal{G} : (Y, \mathrm{RC}_3(Y)) \to (D, X),$
Reverse mode: $\quad \mathcal{G}^{-1} : (D, X) \to (Y, \mathrm{RC}_3(Y)).$

The invertibility of $\mathcal{G}$ is achieved by its invertible structure, containing three encoder blocks (EBs) and decoder blocks (DBs), each with coupling layers for invertible feature processing and invertible

scaling layers for multi-scale representation. Each EB/ DB has dual paths for processing latent and degradation features, with split-concatenation short paths between them for multi-scale feature integration. Conceptually, each EB/DB in AMSIN can be viewed an iterative step for separating latent image and degradation map in a fracture space, with the embedded perfect reconstruction constraint fulfilled by the network's invertibility.

The invertible downscaling (upscaling) operations on the two paths are defined by two different pairs of operations: (a) pixel unshuffle (PUS) and pixel shuffle (PS); and (b) Haar transform (HT) and the inverse HT (IHT). This scheme forms asymmetric dual paths, enhancing spatial and frequency analysis, with high-frequency emphasis given to latent image layer using Haar-wavelet-based features. In addition, an asymmetrically designed reconstruction loss at various scales aids in gradient propagation and fits the asymmetric input and structure of AMSIN.

### 3.2 Detailed Structures

***Coupling layers:*** Coupling layers, as basic invertible units in AMSIN, enable feature interaction and information exchange across dual paths. These layers can operate in the forward or reverse modes. In the forward mode, they transform an input pair $(A_1, A_2)$ to an output pair $(B_1, B_2)$ of identical dimensions, using the following scheme:

$$B_1 = A_1 + \mathcal{F}(A_2), \quad (1)$$
$$B_2 = A_2 \odot \exp(\mathcal{G}(B_1)) + \mathcal{H}(B_1), \quad (2)$$

where $\odot$ denotes element-wise multiplication, and $\mathcal{F}, \mathcal{G}, \mathcal{H}$ are learnable modules implemented as residual blocks with channel attention [56]. In the reverse mode, $(B_1, B_2)$ is perfectly mapped back to $(A_1, A_2)$ through the process:

$$A_2 = (B_2 - \mathcal{H}(B_1)) \oslash \exp(\mathcal{G}(B_1)), \quad (3)$$
$$A_1 = B_1 - \mathcal{F}(A_2), \quad (4)$$

where $\oslash$ denotes element-wise division.

***Encoder and decoder blocks:*** Leveraging coupling layers, each EB and DB keep dual paths for processing. In the forward pass, these paths disentangle distortion map features from latent image features, extracting a distortion-free latent image in the end. For multi-scale processing, each EB initially down-scales dual-path features to the current scale using invertible downscaling, then disentangles and interacts features across paths with coupling layers. Each DB processes features at the current scale via coupling layers and up-scales them to the next level with invertible upscaling. The downscaling, upscaling, and coupling layers ensure that all EBs and DBs maintain invertibility.

***Invertible downscaling and upscaling:*** Two pairs of downscaling and upscaling operations are used in AMSIN. The first pair is (PUS, PS), being inverse to each other. The PUS rearranges adjacent pixels within a 2×2 window into four downscaled images with half resolution, and the PS reassembles these images into the original data. The second pair is (HT, IHT), also functioning as complementary operations. The HT reduces input size using stride 2 convolution with kernels $LL^\top, HL^\top, LH^\top, HH^\top$, where $L = \frac{1}{\sqrt{2}}[1, 1]^\top$ and $H = \frac{1}{\sqrt{2}}[1, -1]^\top$. The $LL^\top$ acts as average pooling, while $HL^\top, LH^\top, HH^\top$ capture edge details (high frequencies).

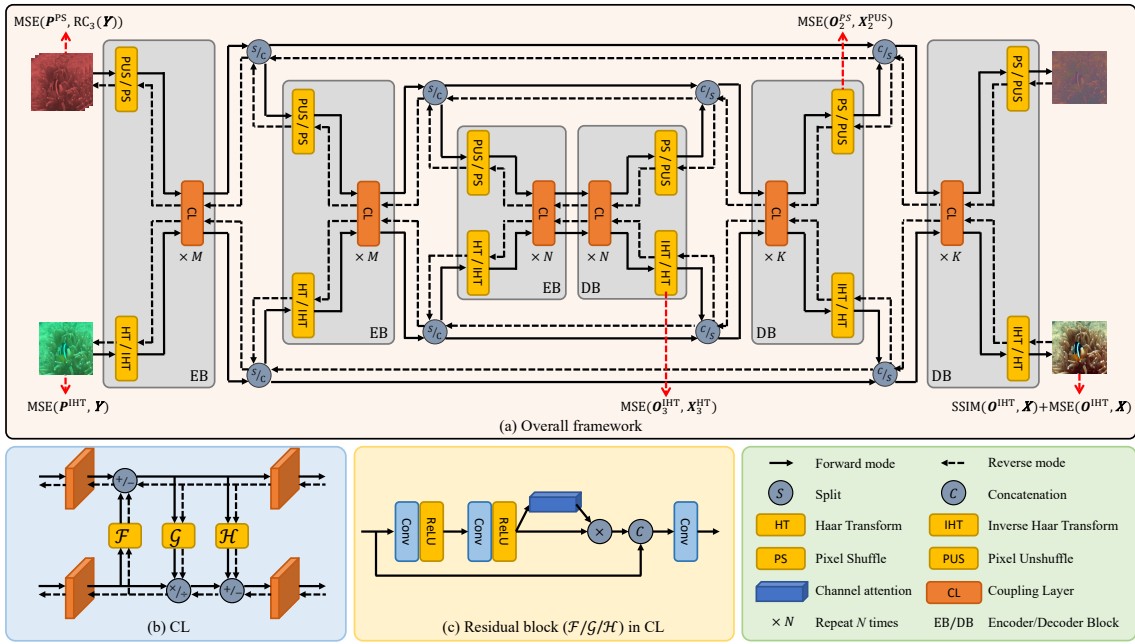

**Figure 1: Architecture of proposed AMSIN. Operations before (after) "/" indicate those used in forward (reverse) modes.**

The IHT reconstructs original data using transposed convolution with these kernels.

These two pairs of operations facilitate multi-scale analysis by reducing the input's size. The key distinction is that the PUS yields spatial content, whereas the HT generates frequency content. Consequently, the PUS-based path excels in spatial analysis, while the HT-based path is more attuned to frequency analysis, enhancing image detail restoration. This asymmetric design facilitates the separation between latent image features and degradation features.

***Split-concatenation short paths:*** Skip connections are vital for the performance of a CNN, but adding them to an INN can compromise its invertibility. To facilitate feature transmission during multi-scale analysis in AMSIN while preserving invertibility, we use split and concatenation operations for short paths between the middle EBs and DBs. On the encoder side, the EB's outputs are divided: one part is downscaled and passed to the next EB, while the other is sent to the corresponding DB. On the decoder side, the previous DB's output is upscaled and merged with features from the corresponding EB for the next DB. The split ratio is set to 1:1 for the PUS/PS-based path and 1:3 for the HT/IHT-based path.

## 3.3 Loss Function

Though the GTs of degradation map $D$ are always unavailable, supervision on $X$ suffices as $D$ is complementary to $X$ in the invertible AMSIN. The invertibility also allows using both the forward and reverse modes for loss calculation:

$$\mathcal{L}_{\text{total}} := \mathcal{L}_{\text{forward}} + \alpha \mathcal{L}_{\text{reverse}}, \quad \alpha \in \mathbb{R}^+. \tag{5}$$

Let $O_i^{\text{PS}}$ and $O_i^{\text{IHT}}$ denote the output of the $i$th DB at the PS-based and IHT-based paths, respectively. The final output, denoted by $(O^{\text{PS}}, O^{\text{IHT}})$, is the output pair of the last DB. Let $X_i^{\text{PUS}}$ and $X_i^{\text{HT}}$

denote the GT image downscaled to the scale of the $i$th DB via PUS and HT, respectively. The forward loss is then defined by

$$\mathcal{L}_{\text{forward}} := \text{SSIM}(O^{\text{IHT}}, X) + \lambda_1 \text{MSE}(O^{\text{IHT}}, X)$$
$$+ \lambda_2 \text{MSE}(O_2^{\text{PS}}, X_2^{\text{PUS}}) + \lambda_3 \text{MSE}(O_3^{\text{IHT}}, X_3^{\text{HT}}), \tag{6}$$

where $\lambda_1, \lambda_2, \lambda_3 \in \mathbb{R}^+$, MSE denotes the mean squared error, and SSIM denotes the structural similarity loss. In implementation, $O_2^{\text{PS}}$ has 6 channels while $X_2^{\text{PUS}}$ contains 12 (4× RGB). Three blue and green channels are discarded from $X_2^{\text{PUS}}$ for dimension consistency, and for emphasizing the recovery on red channel, corresponding to the red-channel replicates in the input for the PUS/PS-based path.

REMARK 1. *A simpler definition of $\mathcal{L}_{forward}$ involves replacing $MSE(O_2^{PS}, X_2^{PUS})$ with $MSE(O_2^{IHT}, X_2^{HT})$. However, (6) is used as it improves gradient flow along the PUS/PS-based path in training and encourages the NN to exploit both paths with distinct downscaling/upscaling schemes for latent image prediction. We cannot supervise both paths at the same scale with GT latent images, as AMSIN predicts both latent images and degradation maps.*

The reverse loss in AMSIN is defined using a data substitution strategy. For an output pair $(O^{\text{PS}}, O^{\text{IHT}})$ from AMSIN where $O^{\text{PS}}$ denotes predicted distortion map and $O^{\text{IHT}}$ denotes the predicted latent image, we replace $O^{\text{IHT}}$ by the GT $X$ and feed $(O^{\text{PS}}, X)$ to the AMSIN in a reverse mode (*i.e.*, from the right to the left in Fig. 1), resulting a pair $(P^{\text{PS}}, P^{\text{IHT}})$. Then, the reverse loss is defined by

$$\mathcal{L}_{\text{reverse}} := \text{MSE}(P^{\text{PS}}, \text{RC}_3(Y)) + \text{MSE}(P^{\text{IHT}}, Y). \tag{7}$$

As the invertible structure of AMSIN has already ensured the perfect reconstruction property of the decomposition process, the reverse loss acts as a data augmentation for regularization.

## 4 Experiments

### 4.1 Experimental Settings

***Datasets:*** Six benchmark datasets are utilized for performance evaluation in our experiments: including UIEBD [30], SUD [29], UCCS [40], EUVP [24], SUIM [23], and U45 [34]. The UIEBD dataset is composed of two subsets: (a) UIEBD-P consisting of 890 raw real underwater images paired with corresponding high-quality reference images as GTs; and (b) UIEBD-UP consisting 60 challenging real underwater images without references nor GTs. The SUD is a synthesized dataset consisting of 10 different types, each characterized by different attenuation coefficients [7]. There are 130 paired images in each type. The UCCS dataset contains three subsets categorized into bluish, greenish, and blue-green tones, each containing 100 real underwater images. For the EUVP dataset, its validation set is used, comprising 330 real underwater images. The SUIM is a dataset for semantic segmentation of underwater images. Its test set containing 110 real underwater images is used. The U45 [34] dataset contains 45 real underwater images for test.

The experimental configurations of training data on these datasets vary in existing literature. Following [41], we train models using the training set of UIEBD. The test set of UIEBD is composed of two parts: UIEBD-P with paired data and UIEBD-UP with unpaired data. To evaluate the generalization performance, the UIEBD-trained models are also tested on four datasets: EUVP (validation set), UCCS, SUIM (test set) and U45. As for SUD, 100 (30) images of each type are randomly selected for training (test).

***Metrics and methods for comparison:*** When GTs are available, we quantify performance using PSNR and SSIM, two standard full-reference metrics. Otherwise, we adopt two non-reference metrics tailored for underwater images quality assessment: UIQM [45] and UCIQE [63]. For performance comparison, we choose (a) two traditional non-learning methods: EUIVF [4] and AACP [57]; and (b) ten deep learning-based UIE methods: Water-Net [30], FGAN [24], SGUIENet [49], TACL [41], USFormer [47], TrinityNet [9], NU$^2$Net [16], GUPDM [44], DM-Water [54], Semi-UIR [21]. We also compare with a representative INN of image processing, InvDN [42], increasing its coupling layers to match our model size. Whenever applicable, we quote the results of these methods from existing literature; otherwise, we retrain them using the same data as ours. For Semi-UIR, the unpaired set of EUVP is also utilized in its retraining.

***Implementation details of AMSIN:*** Through all experiments, we set the number of coupling layers in AMSIN as follows: 3, 3, 2 for the encoder blocks, and 2, 8, 8 for the decoder blocks. For the parameter of loss function, we set $\alpha$ to 0.5, and set $\lambda_1, \lambda_2, \lambda_3$ as 4, 2, 1, respectively. In training, the model weights are initialized by the Xavier method. The Adam optimizer is called with batch size 1. The initial learning rate is set to $2e^{-4}$ for the first 600 epochs and $2e^{-5}$ for the last 50 epochs. Our code is written in PyTorch and run on an NVIDIA GTX 4090 GPU, which is released on GitHub.

### 4.2 Results and Analysis

***Quantitative comparison in terms of full-reference metrics:*** Table 1 summarizes the PSNR and SSIM results on two GT-available datasets, UIEBD-P and SUD. We can see that our AMSIN is the best performer across all the datasets with respect to both PSNR and

**Table 1: Quantitative comparison in full-reference metrics on two benchmark datasets. Bold: best; and Underline: 2nd-best.**

| Method | Source | UIEBD-P PSNR(dB)/SSIM | SUD PSNR/SSIM |
|---|---|---|---|
| EUIVF [4] | CVPR2012 | 17.59/0.787 | 13.15/0.712 |
| AACP [57] | TCSI2017 | 18.51/0.795 | 14.01/0.688 |
| WaterNet [30] | TIP2019 | 20.44/0.852 | 19.03/0.835 |
| FGAN [24] | RAL2020 | 18.52/0.811 | 17.95/0.719 |
| InvDN [42] | CVPR2021 | 19.71/0.778 | 20.81/0.792 |
| SGUIENet [49] | TIP2022 | 23.08/0.895 | 22.52/0.832 |
| TACL [41] | TIP2022 | 22.30/0.888 | 16.97/0.701 |
| USFormer [47] | TIP2023 | 22.01/0.893 | 22.19/0.845 |
| TrinityNet [9] | TGRS2023 | 21.58/0.891 | 21.31/0.837 |
| NU$^2$Net [16] | AAAI2023 | 22.99/0.899 | 22.28/0.844 |
| GUPDM [44] | ACMMM2023 | 22.15/0.889 | 21.80/0.851 |
| DM-Water [54] | ACMMM2023 | 23.52/0.907 | 23.24/0.878 |
| Semi-UIR [21] | CVPR2023 | 23.31/0.897 | 22.37/0.861 |
| AMSIN | Proposed | **24.16/0.918** | **24.70/0.912** |

SSIM. Impressively, AMSIN outperforms other compared methods by a significant margin on SUD, a synthetic dataset. Moreover, on UIEBD-P which is a real-world dataset, our AMSIN also surpasses the second-best performer noticeably, with 0.64dB improvement in PSNR. These results on two types of datasets highlight the exceptional capabilities of AMSIN. Specifically, AMSIN shows significant superiority over the InvDN that has a representative INN structure, demonstrating the effectiveness of our NN architecture design.

***Quantitative comparison in terms of no-reference metrics:*** The quantitative results of different methods on the six datasets in terms of UIQM and UCIQE are listed in Table 2. It can be observed that AMSIN achieves the best UCIQE results on 5/6 datasets, demonstrating its superiority over existing methods, particularly in terms of chroma, saturation, and contrast, measured by the UCIQE metric. In addition, AMSIN ranks among the top two on three datasets in terms of UIQM measuring colorfulness, sharpness, and contrast. Its UIQM results are worse than that of Semi-UIR, a semi-supervised method exploiting additional unpaired training data. Even that, AMSIN is the best performer overall.

***Qualitative comparison via visual inspection:*** The qualitative comparison results are shown in Fig. 2 and Fig. 3. It can be seen that AMSIN also outperforms other compared methods in terms of visual quality. For instance, as seen in Fig. 2, AMSIN excels in handling images with a bluish or greenish tint, resulting in superior visual effects. In contrast, the WaterNet removes the green effect excessively or exhibits residual haze effects; see *e.g.*, the green moss in the first sample or the residual yellowish haze in the second sample; the NU$^2$Net showcases a relatively weak color correction effect; and the SemiUIR tends to excessively eliminate the green effect and haze. AMSIN also demonstrates advantages in preserving color contrast and details. As seen in Fig. 3, AMSIN effectively corrects the color distortion on the rocks and retains the intricate texture details in the second sample. In the third sample, AMSIN removes the green tone caused by water while retaining the natural green hues of the fish and the grass on the ground.

**Table 2: Quantitative comparison in non-reference metrics on six datasets. Bold: best; and Underline: 2nd-best.**

| Method | Source | UIEBD-P | | UIEBD-UP | | EUVP | | UCCS | | SUIM | | U45 | |
|---|---|---|---|---|---|---|---|---|---|---|---|---|---|
| | | UIQM | UCIQE | UIQM | UCIQE | UIQM | UCIQE | UIQM | UCIQE | UIQM | UCIQE | UIQM | UCIQE |
| EUIVF [4] | CVPR2012 | 3.0943 | 0.6261 | 2.1900 | 0.5592 | 3.1790 | 0.5924 | 3.5596 | 0.5683 | 2.3092 | 0.6011 | 4.0223 | 0.5926 |
| AACP [57] | TCSI2017 | 3.1455 | 0.6145 | 2.2975 | 0.5914 | 3.1351 | 0.6120 | 3.9469 | 0.5453 | 2.2831 | 0.6129 | 4.2553 | 0.6039 |
| WaterNet [30] | TIP2019 | 3.2983 | 0.5680 | 2.1424 | 0.5509 | 3.2363 | 0.5584 | 3.2739 | 0.5348 | **2.6614** | 0.5809 | 4.3102 | 0.5553 |
| FGAN [24] | RAL2020 | 3.2279 | 0.5857 | 2.1868 | 0.5488 | 3.1648 | 0.5618 | 3.7785 | 0.5436 | 2.3681 | 0.5941 | 4.2799 | 0.5976 |
| InvDN [42] | CVPR2021 | 3.4105 | 0.5818 | 2.2159 | 0.5440 | 3.0982 | 0.5539 | 3.8111 | 0.5054 | 2.2864 | 0.5958 | 4.1941 | 0.5553 |
| SGUIENet [49] | TIP2022 | 3.3121 | 0.6054 | 2.2448 | 0.5845 | 3.2251 | 0.5850 | 3.3471 | 0.5352 | 2.4409 | 0.6148 | 4.3736 | 0.6148 |
| TACL [41] | TIP2022 | 3.3320 | 0.6101 | 2.4966 | 0.5770 | 3.1365 | 0.6065 | 4.1133 | **0.5826** | 2.5011 | 0.6150 | 4.3910 | 0.6198 |
| USFormer [47] | TIP2023 | 3.2925 | 0.5986 | 2.1317 | 0.5658 | 3.1294 | 0.5862 | 4.0145 | 0.5458 | 2.3381 | 0.5983 | 4.3898 | 0.5886 |
| TrinityNet [9] | TGRS2023 | 3.4078 | 0.5845 | 2.1640 | 0.5459 | 3.1544 | 0.5607 | 3.4228 | 0.5140 | 2.2825 | 0.6006 | 4.2119 | 0.5695 |
| NU$^2$Net [16] | AAAI2023 | 3.4062 | 0.5937 | 2.2687 | 0.5652 | **3.5390** | 0.5773 | 4.1407 | 0.5388 | 2.4702 | 0.5989 | 4.4064 | 0.5931 |
| GUPDM [44] | ACMMM2023 | 3.3814 | 0.5902 | 2.2249 | 0.5809 | 3.2474 | 0.5815 | 3.9659 | 0.5392 | 2.3517 | 0.6059 | 4.2360 | 0.6021 |
| DM-Water [54] | ACMMM2023 | 3.5183 | 0.6223 | 2.1701 | 0.5976 | 3.2009 | 0.6012 | 4.1286 | 0.5591 | 2.4439 | 0.6153 | 4.3552 | 0.6188 |
| Semi-UIR [21] | CVPR2023 | 3.5919 | 0.6170 | **2.5300** | 0.5877 | 3.1408 | 0.6092 | **4.3669** | 0.5530 | 2.5715 | 0.6232 | **4.5150** | 0.6182 |
| AMSIN | Proposed | **3.6040** | **0.6308** | 2.2843 | **0.6081** | 3.2115 | **0.6141** | 4.1493 | 0.5780 | 2.3758 | **0.6348** | 4.4598 | **0.6216** |

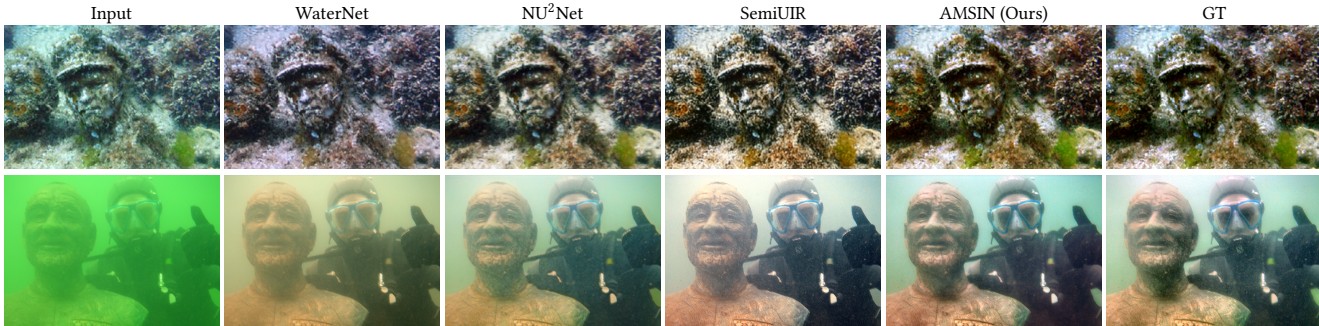

| Input | WaterNet | NU$^2$Net | SemiUIR | AMSIN (Ours) | GT |

**Figure 2: Visual comparison of UIEBD-P.**

**Table 3: Complexity results of different methods.**

| Method | Source | #Params(M) | #MACs(G) | Time(s) |
|---|---|---|---|---|
| Waternet | TIP2019 | 1.09 | 285.91 | 0.002 |
| FGAN | RA-L2020 | 7.02 | 40.96 | 0.001 |
| InvDN | CVPR2021 | 4.95 | 356.54 | 0.020 |
| SGUIENet | TIP2022 | 18.63 | 693.6 | 0.153 |
| TACL | TIP2022 | 11.38 | 227.72 | 0.007 |
| USFormer | TIP2023 | 65.60 | 132.4 | 0.071 |
| TrinityNet | TGRS2023 | 20.14 | 123.89 | 0.048 |
| NU$^2$Net | AAAI2023 | 3.15 | 41.92 | 0.004 |
| GUPDM | ACMMM2023 | 1.49 | 383.98 | 0.121 |
| DM-Water | ACMMM2023 | 10.69 | 534.83 | 0.275 |
| Semi-UIR | CVPR2023 | 1.67 | 145.75 | 0.029 |
| AMSIN | Proposed | 4.63 | 148.88 | 0.012 |

**Table 4: Results of ablation studies on UIEBD-P.**

| Model Setting | PSNR(dB) | Model Setting | PSNR(dB) |
|---|---|---|---|
| Original model | 24.16 | - | - |
| (a) Single scale | 22.95 | (f) Single-path $\mathcal{L}_{forward}$ | 23.81 |
| (b) w/o short path | 23.33 | (g) w/o SSIM loss | 23.99 |
| (c) w/o RC emphasis | 23.91 | (h) w/o $\mathcal{L}_{reverse}$ | 23.96 |
| (d) Only HT/IHT | 23.62 | (i) w/o Coupling | 22.15 |
| (e) Only PUS/PS | 23.51 | (j) Fully non-invertible | 21.06 |

**Complexity comparison:** We compare the complexity of different DNN models in terms of the number of parameters, the number of multiply-accumulate operations (#MACs), and the running time to process a $512 \times 512$ image. See Table 3 for the results. The AMSIN outperforms its top competitor, Semi-UIR, in terms of running time. It is also comparable to Semi-UIR in terms of the model size and #MACs. Therefore, we can conclude that the performance gain of

AMSIN is mainly from its efficient architecture, without additionally introducing noticeable computational complexity. All above results have shown the advantages of our method in both underwater image enhancement performance and computational complexity.

**Visualization of estimated distortion maps:** See Fig. 4 for the predicted degradation maps and the re-degraded images formed by GTs and predicted degradation maps. Obviously, we can reconstruct re-degraded images that closely match the input degraded images. This observation highlights the capability of the predicted distortion map to accurately re-produce the true distortion effects.

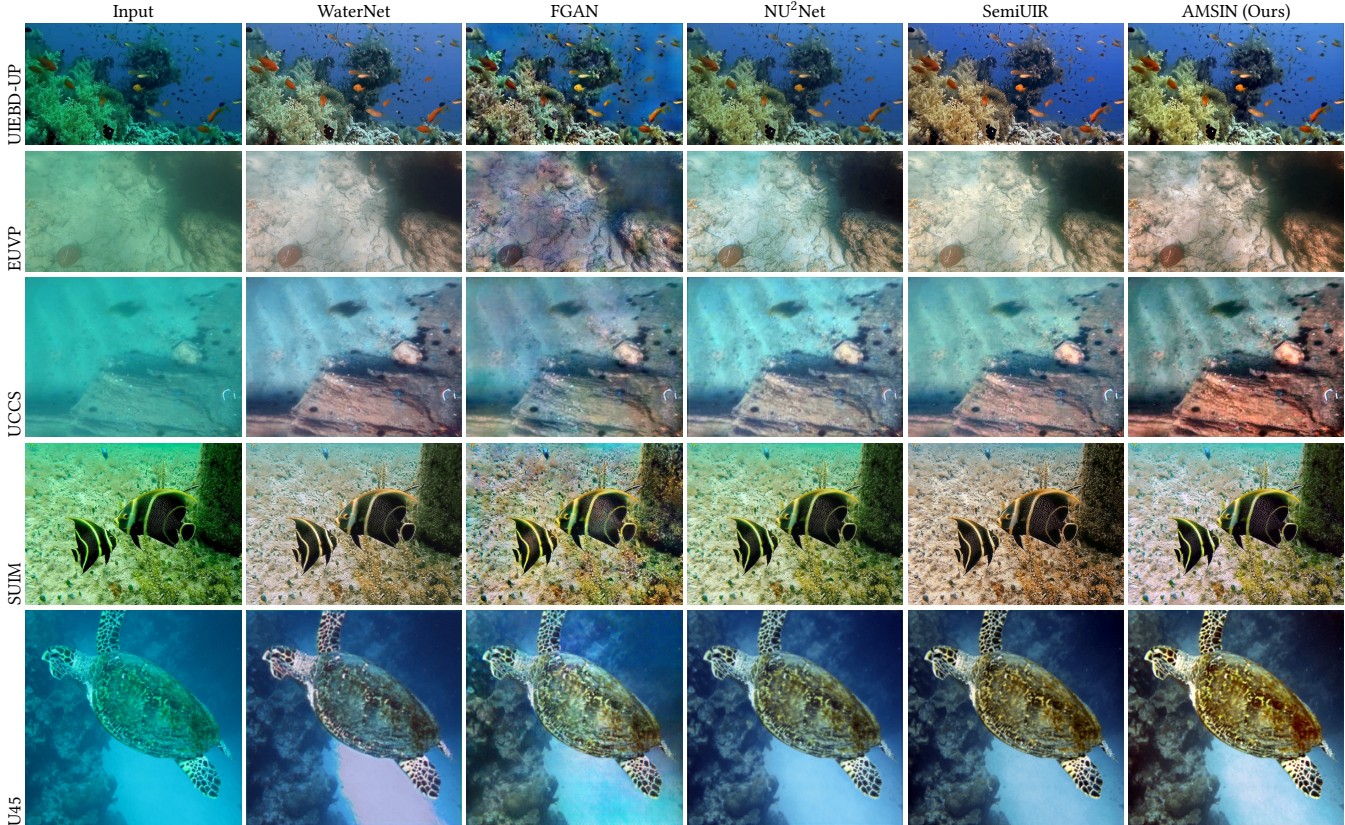

**Figure 3: Visual comparison of enhanced images by different methods.**

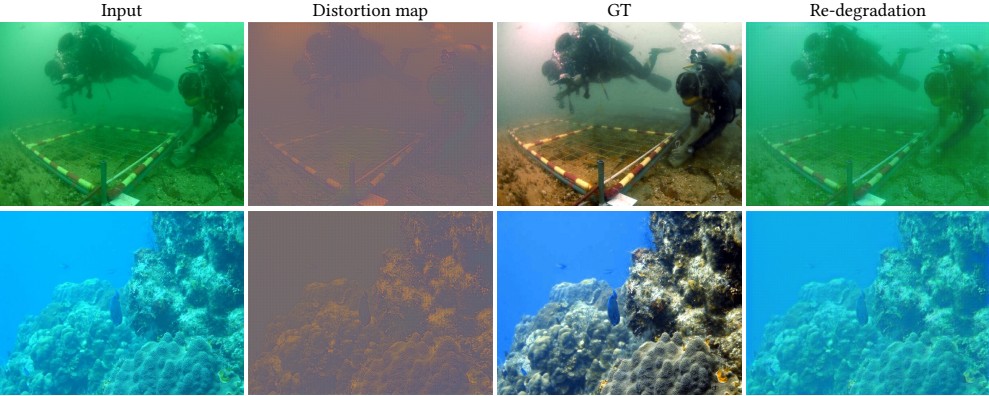

**Figure 4: Visualization of distortion maps and re-degraded images. The re-degradation is done by composing the estimated distortion map and the GT back to a degraded image, using reverse mode of our INN.**

## 4.3 Ablation Studies

To see the performance contribution of each of its key components, we conduct ablation studies by forming the following baseline models from AMSIN. (a) **"Single scale"**: All scaling operations in AMSIN are discarded. (b) **"w/o short path"**: All split-concatenation short paths are removed from AMSIN. (c) **"w/o RC emphasis"**: It replicates the whole degraded image rather than the red channel, for the AMSIN's input, and replace the red-channel emphasized $X_2^{\text{PUS}}$ by a RGB channel-uniform one in the loss $\mathcal{L}_{\text{forward}}$. (d) **"Only HT/IHT"**: All PUS/PS operations are replaced with HT/IHT to form symmetric dual paths. (e) **"Only PUS/PS"**: All HT/IHT operations are replaced with PUS/PS to form symmetric dual paths. (f) **"Single-path $\mathcal{L}_{\text{forward}}$"**: It replaces $\text{MSE}(O_2^{\text{PS}}, X_2^{\text{PUS}})$ with $\text{MSE}(O_2^{\text{IHT}}, X_2^{\text{HT}})$ in $\mathcal{L}_{\text{forward}}$. (g) **"w/o SSIM**

| Input | SGUIENet | TACL | NU²Net | SemiUIR | AMSIN (Ours) |

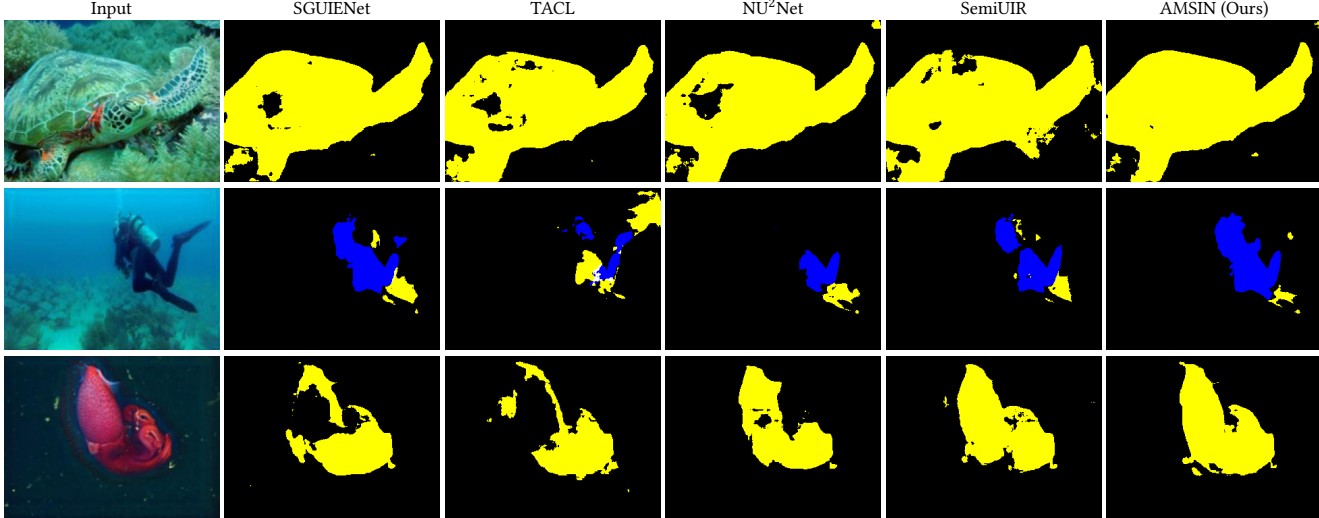

**Figure 5: Visual results of semantic segmentation on UIE results. Input images are from the the EUVP dataset [24].**

**loss"**: It excludes the SSIM loss from $\mathcal{L}_{\text{forward}}$. (h) **"w/o $\mathcal{L}_{\text{reverse}}$"**: It disables the reverse loss during training. (i) **"w/o Coupling"**: All coupling layers are replaced by three residual blocks same as those contained in the coupling layers. As a result, the EBs and DBs in AMSIN are now non-invertible. (j) **"Fully non-invertible"**: Based on above, the invertible scaling operators are further replaced by the standard non-invertible ones done within the convolutional layers, and the split concatenation short paths are replaced by the common skip connections with addition.

See Table 4 for the results of the ablation studies. The original AMSIN outperforms all the baselines noticeably, demonstrating the effectiveness of the key components in AMSIN. We provide a detailed analysis as follows. (a) A significant PSNR decrease occurs when using a single-scale structure, emphasizing the significant role of the multi-scale structure. (b) A noticeable PSNR drop when removing the split-concatenation short paths, demonstrating their necessity. (c) Using asymmetric input leads to certain performance gain, as it enhances the recovery of red-channel information. (d)&(e) The asymmetric downscaling/upscaling in the dual paths is important for the performance. Purely using PUS/PU or HT/IHT yields sub-optimal results. (f) Without incorporating the asymmetric loss, a noticeable PSNR loss is observed. (g)&(h) The SSIM loss and reverse loss have certain contribution to the PSNR gain, though not big. This is because the invertible structure of AMSIN already has strong regularization to avoid overfitting. (i)&(j) These two ablation studies verified the usefulness of introducing invertibility into AMSIN. Without the invertibility, AMSIN shows a noticeable performance drop, demonstrating the importance of utilizing an invertible NN structure for decomposition-based UIE. To conclude, each proposed component in our AMSIN has a noticeable contribution to the performance.

### 4.4 Evaluation on Downstream Segmentation

To further verify the practicability of our AMSIN for subsequent downstream tasks, we apply the semantic segmentation method [23]

**Table 5: Segmentation performance on underwater images enhanced by different UIE methods, in terms of two metrics.**

| Object | Metric | SGUIENet | TACL | NU2Net | SemiUIR | AMSIN |
|--------|--------|----------|------|--------|---------|-------|
| HD | $\mathcal{F}$-score | 82.15 | 78.40 | 79.31 | 81.31 | **86.02** |
| | mIOU | 72.20 | 69.44 | 71.31 | 70.91 | **75.22** |
| FV | $\mathcal{F}$-score | 85.82 | 88.87 | 87.57 | 87.05 | **88.93** |
| | mIOU | 77.83 | **79.98** | 78.68 | 77.89 | 79.28 |

to the UIE results of different methods and then compare the segmentation accuracy. The top competitors in previous experiments are selected for comparison. See Fig. 5 for the segmentation results on some enhanced images. We can see that the objects segmented using the images enhanced by AMSIN are more complete than that of other compared methods. In addition, we also evaluate the segmentation accuracy on the categories of Human divers (HD) and Fish and vertebrates (FV) on the SUIM dateset utilizing two metrics used in [23]: $\mathcal{F}$-score and mIOU. See Table 5 for the result. The AMSIN performs the best in 3/4 cases, demonstrating that our approach not only improves visual quality, but also benefits downstream visual tasks.

## 5 Conclusion

This paper addressed the challenges in UIE by proposing a novel NN architecture. Treating UIE as a decomposition problem, our NN utilizes a multi-scale invertible structure to maintain reconstruction constraints and extract cross-scale cues during the decomposition process. Additionally, we introduced an asymmetric dual-flow scaling design for enhanced spatial and frequency analysis. We also incorporated an asymmetric input form to enhance the recovery of red-channel information, with an asymmetric multi-scale reconstruction loss introduced to improve model training. Extensive experiments have demonstrated the superior performance of our approach compared to existing ones.

## Acknowledgments

This work is supported by Science and Technology Plan Project of Guangzhou under Grants 2023A04J1681 and 2024B01W0007, Fundamental Research Funds for Central Universities under Grant x2jsD2230220, Natural Science Foundation of Guangdong Province under Grants 2023A1515012841 and 2022A1515011755, National Natural Science Foundation of China under Grants 62372186 and 62072188, National Foreign Expert Project of Ministry of Science and Technology of China under Grant G2023163015L, National Key Research and Development Program of China under Grant 2024YFE0105400, and Singapore MOE AcRF Tier 1 under Grant A-8000981-00-00.

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
