# OpenReview forum: "Enhancing Underwater Images via Asymmetric Multi-Scale Invertible Networks"
_acmmm.org/ACMMM/2024/Conference — MM2024 Poster_

### Official Review · Reviewer_1fdt · 2024-05-23

**Rating:** 4
**Confidence:** 3

**Summary:**

The overall structure of the article is encoder-decoder, proposing a bidirectional invertible structure Neural network (INN). In the decomposition process, regularization has been added to improve the reconstruction effect. In addition, the multi-scale structure is also used to handle diverse scales of degradation factors in underwater images.

**Strengths:**

This paper regards the underwater image enhancement problem as an image decomposition problem. This method divides underwater images into destruction-free images and destruction maps.
During the decomposition process, this paper focuses on Implicit regulation and multi-scale underwater degradation.

**Limitations:**

1. The poor comparison of images and text in the article leads to a decrease in readability and affects overall understanding.
2. The comparison of this asymmetric with the replacement of HT/IHT and PUS/PS is not shown.
3. More comparisons with the model-based and fusion-based are needed to demonstrate the superiority of performance, like
[A] Wang et al.: “Single Underwater Image Restoration Using Adaptive Attenuation-Curve Prior,” IEEE Transactions on Circuits and Systems I: Regular Papers, 2018.
[B] Ancuti et al., “Enhancing underwater images and videos by fusion,” in Proc. of IEEE Int. Conf. Comput. Vis. Pattern Rec. (CVPR), 2012.
4. Error: "denotes element-wise division," on Line 311, should be multiplication.

**Suitability:**

2

---

### Official Review · Reviewer_1EFe · 2024-05-24

**Rating:** 2
**Confidence:** 4

**Summary:**

This paper addresses the underwater image enhancement problem by proposing a novel method, namely Asymmetric Multi-Scale Invertible Network (AMSIN). The authors redefine underwater image enhancement as an image decomposition problem and design a deep invertible neural network to predict the latent clear image and the degradation map. The network leverages a multi-scale structure and several asymmetric design elements optimized for the physics of underwater imaging. Experiments show that the method achieves significant performance improvements compared to existing methods.

**Strengths:**

1. Utilizes an invertible network to implement the decomposition process, providing necessary regularization through the inherent reconstruction constraint.

2. Incorporates the red channel information as an additional input to the network and supervising it during the training process.

3. Integrates several asymmetric design elements optimized for the decomposition model and the physics of underwater imaging.

**Limitations:**

1. Lack of novelty in the decomposition problem formulation. The decomposition-based approach has been widely used in various low-level vision tasks, such as super-resolution and image restoration. The authors' claim of innovatively redefining underwater image enhancement as a decomposition problem may not be as novel as presented, given its widespread adoption in other related fields.
2. Limited innovation in the multi-scale network structure. The proposed AMSIN model appears to be a modified version of the well-established U-Net architecture. While the authors introduce some asymmetric design elements and invertible components, the core structure of the network is not significantly different from the standard U-Net.
3. Limited comparison with state-of-the-art methods. The experimental section does not include a comprehensive comparison with the latest and most advanced **dataset** and **methods** in underwater image enhancement. For example:
    1. U-shape Transformer for Underwater Image Enhancement, TIP2023, also, the **LSUI** dataset in this paper is more convincing than SUD since SUD is synthesized.
    2. A Generalized Physical-knowledge-guided Dynamic Model for Underwater Image Enhancement, MM2023
    3. Underwater Image Enhancement by Transformer-based Diffusion Model with Non-uniform Sampling for Skip Strategy, MM2023
4. The authors' decision not to evaluate full-reference metrics using the EUVP dataset raises questions. It appears that the EUVP dataset includes paired data, making it suitable for full-reference evaluation.
5. While this paper focuses on unimodal image processing—a topic that may not be the primary interest of the MM community—the authors can demonstrate the method's value by applying it to underwater video enhancement, thus proving its relevance in multimedia/multimodal processing.

**Suitability:**

1

---

### Official Review · Reviewer_wLw4 · 2024-05-25

**Rating:** 4
**Confidence:** 3

**Summary:**

The paper redefines underwater image enhancement as an image decomposition problem and proposes a deep invertible neural network. The INN is structured on a multi-scale basis to effectively manage the varied scales of degradation factors and incorporates several asymmetric design elements that are specifically optimized for the decomposition model and the unique physics of underwater imaging.

**Strengths:**

1.The ablation experiments are conducted comprehensively, and methods were evaluated across multiple datasets.

2.This method appears to present more details and reduce color distortion.

3.The proposed framework appears to perform well in downstream segmentation tasks.

**Limitations:**

1.From line 121 to line 124, more references are needed to support the explanation for the overfitting issue.

2.The annotations for some operations in Figure 2 are not clear enough, such as the lack of explanations for S/C and C/S, making it less readable.

3.For the split and concatenation operation, you set a 1:3 ratio for the HT/IHT-based path. After the Haar wavelet transform, the four components actually represent different aspects (vertical, horizontal, etc.). How do you choose which channels to send to the EB and DB?

4.For different components of $L_{forward}$, how you decide the ratio of $λ1$, $λ2$, $λ3$?

**Suitability:**

2

---

### Meta-Review · Area_Chair_HTXQ · 2024-06-28

**Recommendation:** Accept (Poster)
**Confidence:** 5

**Metareview:**

This paper addresses the underwater image enhancement problem by proposing a novel framework, the Asymmetric Multi-Scale Invertible Network (AMSIN). The paper received mixed ratings from the reviewers.

Reviewer wLw4 assigned a Borderline Accept rating, noting that the method was evaluated across multiple datasets, with extensive ablation studies conducted. The proposed framework performs well in downstream segmentation tasks.

Although the rebuttal addressed the major concerns of Reviewer 1EFe, he (or she) believes the paper lacks an evaluation using full-reference metrics on the established EUVP dataset, and thus recommends rejecting the paper.

Reviewer 1fdt leans toward accepting the paper after the rebuttal, holding an opposite opinion to Reviewer 1EFe regarding the necessity of full-reference metrics on the EUVP dataset. Reviewer 1fdt argues that for underwater image restoration, ground-truth images do not exist, and the EUVP dataset's ground truth is generated by CycleGAN, which does not provide true full-reference metrics. Therefore, Reviewer 1fdt considers full-reference experiments on EUVP unnecessary.

The internal discussion did not reach a consensus. After carefully reading the paper, its rebuttal, and the discussion, the AC finds the paper well-written and well-justified by the experiments. The AC agrees with Reviewer 1fdt's perspective that the lack of full-reference experiments on EUVP is not a valid reason to reject the paper. Given the merits of the work, the AC recommends accepting the paper.

The AC urges the authors to revise their paper by taking into account the suggestions from the reviewers to further strengthen the paper.